# Rapid Characterization of Bacterial Lipids with Ambient Ionization Mass Spectrometry for Species Differentiation

**DOI:** 10.3390/molecules27092772

**Published:** 2022-04-26

**Authors:** Hung Su, Zong-Han Jiang, Shu-Fen Chiou, Jentaie Shiea, Deng-Chyang Wu, Sung-Pin Tseng, Shu-Huei Jain, Chung-Yu Chang, Po-Liang Lu

**Affiliations:** 1Department of Chemistry, National Sun Yat-sen University, Kaohsiung 804201, Taiwan; impossible122@yahoo.com.tw; 2Institute of Medical Science and Technology, National Sun Yat-sen University, Kaohsiung 804201, Taiwan; morethejzh77@gmail.com; 3Department of Marine Biotechnology and Resources, National Sun Yat-sen University, Kaohsiung 804201, Taiwan; sfc@mail.nsysu.edu.tw; 4Department of Medicinal and Applied Chemistry, Kaohsiung Medical University, Kaohsiung 807378, Taiwan; 5Research Center for Environmental Medicine, Kaohsiung Medical University, Kaohsiung 807378, Taiwan; 6Division of Gastroenterology, Department of Internal Medicine, Kaohsiung Medical University Hospital, Kaohsiung 807377, Taiwan; dechwu@yahoo.com; 7Department of Medicine, Faculty of Medicine, College of Medicine, Kaohsiung Medical University, Kaohsiung 807378, Taiwan; 8Regenerative Medicine and Cell Therapy Research Center, Kaohsiung Medical University, Kaohsiung 807378, Taiwan; 9Department of Medical Laboratory Science and Biotechnology, Kaohsiung Medical University, Kaohsiung 807378, Taiwan; tsengsp@kmu.edu.tw; 10Department of Laboratory Medicine, Kaohsiung Medical University Hospital, Kaohsiung 807377, Taiwan; cathy30353035@gmail.com; 11Department of Microbiology and Immunology, Kaohsiung Medical University, Kaohsiung 807378, Taiwan; cychang@kmu.edu.tw; 12Division of Infectious Diseases, Department of Internal Medicine, Kaohsiung Medical University Hospital, Kaohsiung Medical University, Kaohsiung 807377, Taiwan; 13College of Medicine, Kaohsiung Medical University, Kaohsiung 807377, Taiwan

**Keywords:** ambient ionization mass spectrometry, thermal desorption–electrospray ionization/mass spectrometry, multivariate statistical analysis, bacterial specie, lipid profile

## Abstract

Ambient ionization mass spectrometry (AIMS) is both labor and time saving and has been proven to be useful for the rapid delineation of trace organic and biological compounds with minimal sample pretreatment. Herein, an analytical platform of probe sampling combined with a thermal desorption–electrospray ionization/mass spectrometry (TD-ESI/MS) and multivariate statistical analysis was developed to rapidly differentiate bacterial species based on the differences in their lipid profiles. For comparison, protein fingerprinting was also performed with matrix-assisted laser desorption ionization time-of-flight (MALDI-TOF) to distinguish these bacterial species. Ten bacterial species, including five Gram-negative and five Gram-positive bacteria, were cultured, and the lipids in the colonies were characterized with TD-ESI/MS. As sample pretreatment was unnecessary, the analysis of the lipids in a bacterial colony growing on a Petri dish was completed within 1 min. The TD-ESI/MS results were further performed by principal component analysis (PCA) and hierarchical cluster analysis (HCA) to assist the classification of the bacteria, and a low relative standard deviation (5.2%) of the total ion current was obtained from repeated analyses of the lipids in a single bacterial colony. The PCA and HCA results indicated that different bacterial species were successfully distinguished by the differences in their lipid profiles as validated by the differences in their protein profiles recorded from the MALDI-TOF analysis. In addition, real-time monitoring of the changes in the specific lipids of a colony with growth time was also achieved with probe sampling and TD-ESI/MS. The developed analytical platform is promising as a useful diagnostic tool by which to rapidly distinguish bacterial species in clinical practice.

## 1. Introduction

The sequence of the 16S rRNA encoding gene has long been utilized as the gold standard for bacterial identification and classification. Since the 16S rRNA sequence is culture independent, this approach is especially useful for identifying fastidious microorganisms [1,2,3]. Unfortunately, the sensitivity and specificity for direct sample applications vary considerably, and such explicit identification is not always successful due to highly similar sequences of the 16S rRNA gene in partial bacterial species [4,5]. Since the 16S rRNA gene cannot identify more than one species simultaneously in polymicrobial samples, the primers are designed to be broad range so the PCR can amplify all bacterial DNA present, including whether it is a relevant pathogen or contamination from the bacteria or the chemical reagents used in PCR processes. This makes it prone to false-positive results due to contamination, pre- or post-sampling [6,7]. Another concern is human DNA present as the dominant DNA in samples such as tissue and biofluids, and the bacterial DNA may be hidden due to the large amount of human DNA that correspondingly decreases the sensitivity of the assay. There may also be problems with primer cross-reactivity and co-amplification of human mitochondrial DNA, which also contains variants of the 16S rRNA gene [8]. Therefore, despite its role as the gold standard, not all bacterial species can be confidently identified by differences in their 16S rRNA sequence. This makes the development of other identification approaches necessary [3,9]. Additionally, the genotypic approach generally requires a complex and time-consuming sample preparation; this is comparably expensive and requires at least several hours for identification. For these reasons, the 16S rRNA sequence is rarely used in routine clinical practices but is mostly applied in reference laboratories; thus, the development of an easy and cost-effective approach for the rapid identification of microorganisms has become particularly important.

Mass spectrometry has gained much attention for its capability of identifying microorganisms due to its intrinsic advantages of automatic operation, fast data acquisition, and high sensitivity and specificity [10,11,12]. Previous literature focused mostly on applying gas chromatography/mass spectrometry (GC/MS) or pyrolysis GC/MS to detect unique biomolecules from bacterial cells for identification, but due to the destructive nature of thermal heating effects in GC/MS, only small molecules and fragments of larger molecules were detected [13,14]. The introduction of liquid chromatography/mass spectrometry (LC/MS) allows for the monitoring of biomolecules as intact complex phospholipid species extracted from bacterial cells [15,16]; unfortunately, complex and time-consuming sample preparation processes are needed for this approach, making it impractical to apply LC/MS to identify microorganisms.

The advent of desorption ionization (DI) techniques such as matrix-assisted laser desorption ionization (MALDI) has provided significant momentum to the identification of microorganisms through their protein profiles [17,18], where MALDI analysis focuses on the proteins of ribosomal origin with mass range between 2 and 20 kDa [19]. With a very short analytical and reporting time, this approach has dramatically reduced the turnover time in clinical microbiology laboratories by about one day; additionally, its consumption costs are much lower than those required in conventional bacterial identification techniques [20].

Bacterial identification using the MALDI approach is now available, and it has been routinely used in many clinical microbiology laboratories in the United States of America, Europe, and Asia, as well as in many other countries. The detection of protein ion signals with MALDI is known to be affected by matrix selection, the sweet spot, and ion suppression effects; however, to compensate for these uncertainties, the identification of bacteria with other biomarkers such as lipids with mass spectrometry should be considered. In light of this, a novel metal oxide laser ionization (MOLI) technique using cerium (Ce) as a catalyst to convert bacterial lipids into systematically available fatty acids has been developed [21], enabling a high accuracy for strain-level identification among nine different strains of three staphylococcus species using their fatty acid profiles [22]. However, the similarity between bacteria and mammalian fatty acids may overlap with the direct analysis of patient specimens [23]. Therefore, discovering other representative markers is necessary to effectively differentiate between bacteria and mammalian cells.

As is generally known, bacteria possess remarkably different phospholipid sets from those of mammalian cells [24,25]. For instance, the lipid composition of a typical nucleated mammalian cell comprises 45–50% phosphatidylcholine (PC), 15–25% phosphatidylethanolamine (PE), 10–15% phosphatidylinositol (PL), 5–10% phosphatidylserine (PS), phosphatidic acid, 5–10% sphingomyelin (SM), 2–5% cardiolipin (CL), <1% phosphatidylglycerol, 2–5% glycosphingolipids, and 10–20% cholesterol [24]. Different phospholipid distributions (e.g., PE, PG, and CL) were observed in the growth in *E. coli*. Hence, different bacteria species should be distinguishable based on the differences of their lipid profiles. Traditionally, the approach for distinguishing bacteria through lipid profiles focuses on the detection of volatile and semi-volatile fatty acids by GC/MS, although the requirements for the tedious sample pretreatment make it impractical to use the GC/MS approach to distinguish bacteria in clinical settings. Many different mass spectrometric techniques, including fast atom bombardment (FAB) [26] and LC-ESI/MS [27], have been used to demonstrate different bacteria own species-specific lipid profiles, but none of these approaches possess robustness or high-throughput capability for the identification of bacteria in a clinical laboratory.

Ambient ionization mass spectrometry (AIMS) is capable of rapidly characterizing chemical and biochemical compounds with minimal or no sample pretreatment, making this approach useful in characterizing lipid biomarkers for bacterial identification in a clinical laboratory, and growing interest in applying AIMS to generate lipid profiles from bacterial species has brought about new aspects by which to distinguish microorganisms through the differences in their lipid profiles. Desorption electrospray ionization (DESI) allows the rapid acquisition of highly reproducible lipid fingerprinting from intact microorganisms, and the application of principal component analysis (PCA) on the data obtained by DESI permits sub-species differentiation [28]. Metabolites collected from bacterial colonies were also directly acquired using imaging mass spectrometry based on the DESI technique [29]. Membrane lipids were detected in real time to monitor the variation in the different physiological conditions in cyanobacterial cells via easy ambient sonic-spray ionization mass spectrometry (EASI) coupled with PCA [30]. Bacterial fatty acids were analyzed by direct analysis in real time mass spectrometry (DART) combined with PCA to effectively differentiate ten different bacterial genera [31]. Rapid evaporative ionization mass spectrometry (REIMS) was also successfully used to differentiate different microorganisms based on their lipid profiles [32,33,34]. However, these AIMS techniques partially require bringing the sample (a Petri dish) to the close vicinity of an ion source, which requires a sample-switching step and makes it impossible to directly analyze a single colony that cannot fit in the space between the ion source and the mass inlet. Moreover, the ion source is easily contaminated, since it is unable to control the amount of the injected analytes using the aforementioned AIMS, even though these techniques have emerged as significant tools facilitating the detection of compounds in recent years. In light of these deficiencies, it is difficult to examine a large number of samples during routine clinical microbiology laboratory tests. In addition, it is also difficult to observe the growth conditions of the bacteria or the phenomena of bacteria–bacteria interaction in real time. Therefore, an effective analytical platform by which to perform high-throughput screening is necessary.

To date, thermal desorption–electrospray ionization/mass spectrometry (TD-ESI/MS) combined with probe sampling has been demonstrated to be a useful AIMS for rapidly detecting various chemical compounds without performing tedious sample pretreatment [35]. Volatile or semi-volatile chemical compounds on different surfaces and in solutions were successfully detected by TD-ESI/MS, such as residual pesticides on fruit and vegetable surfaces [36,37], preservatives, UV filters and photoproducts in cosmetics [38,39], phthalates on the surface of various objects [40,41], illicit drugs in adulterated samples [42], and ingested pesticides and medicine in biofluids [43,44,45,46,47]. Due to its simplicity and speed of operation, TD-ESI/MS might be a practical bacterial identification system in clinical laboratories. In the previous literature, directly characterizing disease biomarkers in biological samples can be briefly classified as two parts including characterizing “specific” biomarkers and “profiles (pattern)” recorded from clinical samples [48]. Therefore, TD-ESI/MS was used to directly characterize lipid profiles in bacterial colonies of different bacterial species based on the similar strategy of profiles characterization. The sample was collected by touching a metallic inoculating probe on a single bacterial colony, and the lipids on the sampling probe were desorbed as the4 probe was inserted into a preheated oven, and then carried by a nitrogen stream into an electrospray plume, where they were ionized via ion molecule reactions with charged solvent species to form analyte ions. The analyte ions were subsequently detected by a mass analyzer attached to the TD-ESI source, and the obtained results were further analyzed with PCA and HCA to assist in distinguishing the bacterial species. For comparison, the MALDI approach was used to characterize the proteins from the extracts of different bacterial cultures for species differentiation.

## 2. Results and Discussion

Since less sample pretreatment is required for TD-ESI/MS, once the sample was collected on a probe, it took less than 1 min to complete the analysis, and even though the time required for the analysis was very short, there were actually several analytical processes involved, including: (1) sampling lipids from a single bacterial colony growing on the agar in a Petri dish with a metallic inoculating probe followed by solvent extraction; (2) thermal desorption of the analytes by inserting the probe into the TD-ESI source; (3) delivery of the desorbed analytes into an ESI plume with nitrogen; (4) ionization of the analytes in the ESI plume; and (5) mass spectrometric detection of the analyte ions. The memory effect caused by the residual sample retained on the metallic inoculating probe was avoided simply by burning the probe with a flame from a handheld butane torch for a few seconds after each analysis. Once this clean-up process was complete, the TD-ESI mass spectra showed no ion signals from any residual samples on the probe. Figure 1 illustrates the analytical processes for the analysis of lipids in a single bacterial colony.

To observe the repeatability of using TD-ESI/MS to detect analyte ions recorded in positive mode in bacterial species, a metallic inoculating probe was used to collect a single colony of *E. coli*, a Gram-negative bacterium. The analysis was repeated for twenty measurements on twenty colonies grown on agar in a Petri dish, and the results of these repeated tests are presented in Figure 2. Although only a trace sample was collected on the probe, TD-ESI/MS was sensitive enough to detect analyte ions in the sample. For this reason, the analysis turned out to be simple and rapid, since the samples could be collected in the culture room with multiple probes and brought to the mass spectrometric lab for analysis. The relative standard deviation (RSD) of total ion current (TIC) of twenty analyses on the analytes collected from single colonies of *E. coli*, with TD-ESI/MS was 5.2% (Figure 2a). Figure 2b demonstrates the mass spectral signals acquired in the positive full scan mode from the analysis of the entire blood agar plate; it includes signals of all the amino acids, fatty acids, and lipids found in the entire blood agar plate. The mass spectral signals of the bacterial colonies of interest are actually the result of subtracting the signals of the blood agar matrix from the full scan signals of the entire blood agar plate. Figure 2c,d show the averaged mass spectra of the first and twentieth tests. As can be seen, both of the mass spectra are quite similar, indicating the reliability of the stability and repeatability of TD-ESI/MS. To further identify the signals in bacterial samples, TD-ESI-MS/MS analyses were performed, and the results are shown in Figure 2e–g. Since lipids have a characteristic fragmentation pattern such as m/z 369 (origin from m/z 386 intact cholesterol), it can be possibly recognized as one kind of lipids. Previous literature has also reported that these ion signals are mainly from phosphatidylglycerols (PGs), phosphatidylethanolamines (PEs), and phosphatidic acids (PAs) [32,33,34]. It should be noted that the analytical techniques used in these studies were either GC/MS or LC/MS/MS [48,49].

To explore the lipid profiles of different bacteria, five Gram-negative and five Gram-positive bacterial species were cultured and subjected to TD-ESI/MS analysis (Table 1). It was found that the analyte ions detected from the Gram-positive bacterial colony (Figure 3a) were at least ten times weaker than those of the Gram-negative bacterial colony (Figure 2c,d). This might be why the RSD (27.16%) for the repeated test of TIC for *E. faecalis* (a Gram-positive bacterium) was much higher than that from the *E. coli* colony (a Gram-negative bacterium). There are several reasons for the detection of less analyte ion signals from the Gram-positive bacterial colony with TD-ESI/MS: (1) the content of Gram-positive bacteria is lower than that of Gram-negative bacteria; (2) Gram-positive bacteria have no outer membrane; and (3) Gram-positive bacteria have a thicker peptidoglycan layer outside the plasma membrane than Gram-negative bacteria. Peptidoglycan is a polymer consisting of sugars and amino acids that forms a mesh-like layer outside the plasma membrane of Gram-positive bacteria. The presence of a thick peptidoglycan layer makes it difficult to release lipids from the inner membrane of Gram-positive bacterial cells [50]. Due to the thick layer of peptidoglycan in the Gram-positive bacteria, the analyte ions on the TD-ESI mass spectrum of the *E. faecalis* colony were complex, but the signal intensity was low, as shown in Figure 3a. It was also difficult to distinguish different Gram-positive bacterial species with a poor-quality TD-ESI mass spectrum; to overcome this problem, a simple analytical approach was developed to efficiently detect analytes with TD-ESI/MS from the Gram-positive bacterial species.

Liquid–liquid extraction (LLE) is a traditional sample pretreatment method used to separate and concentrate lipids from the biological sample. The single colony of *E. faecalis* was collected with a probe, and it was immersed in 100 μL of chloroform/methanol solution in a vial and shaken for 5 s. After this, the extracted organic solution was subjected to TD-ESI/MS analysis. Figure 3b shows the TD-ESI mass spectrum of the analytes extracted from a single colony of *E. faecalis*. Only weak ion signals were detected between m/z 550 and m/z 650, indicating that organic solvent extraction is not an efficient method by which to extract analytes from the colony of Gram-positive bacteria. To increase the extraction efficiency of analytes from the bacteria, acid hydrolysis (5% TFA) followed by solvent extraction was used to disrupt peptidoglycan and increase the release of components from the cell membrane of bacteria. Figure 3c shows the TD-ESI mass spectrum of the analytes extracted from a single colony of *E. faecalis* with a chloroform/methanol solution containing 5% TFA. With the presence of TFA treatment, the analyte ion signals (Figure 3c) were much stronger than those extracted only with the chloroform/methanol solution (Figure 3b), while interference from peptidoglycan was not observed for those compounds that were insoluble in chloroform/methanol. This analytical process was subsequently used to treat the samples collected from a single colony of Gram-positive and Gram-negative bacteria prior to TD-ESI/MS analysis.

To study the lipid composition of different bacterial species, five Gram-negative and five Gram-positive bacteria were cultured first, followed by probe sampling and TD-ESI/MS analysis, and the obtained mass spectra were further analyzed with PCA and HCA. Since the mass scan range was set between m/z 400 and 1000 and the desorption temperature in TD-ESI source was set at 280 °C, the ions from large and polar biomolecules such as ribosomal proteins, peptides, and carbohydrates in the bacteria were not desorbed or detected at all. Figure 4 shows the TD-ESI mass spectra detected in the positive mode from the samples collected from the colonies of ten bacterial species. As can be seen, the predominant ions detected on the TD-ESI mass spectra are primarily intact lipids distributed in the mass range of m/z 600–900. In addition, the lipid ion signals detected on Gram-negative bacterial colonies (Figure 4a–e) were much stronger than those of Gram-positive bacteria (3–25 times) (Figure 4f–j), proving that the peptidoglycan layer on Gram-positive bacteria can partially hinder the release of lipids from the cell membrane even with a simple solvent extraction followed by acid hydrolysis. Moreover, the TD-ESI mass spectra of the Gram-negative and Gram-positive bacteria were also recorded in the negative mode by using TD-ESI/MS; however, there were insufficient lipid ion signals by which to distinguish different bacterial species in this study.

To efficiently distinguish between different bacterial species, the analytical results obtained by TD-ESI/MS were further analyzed with PCA. Figure 5a shows the PCA score plots for ten species of bacteria, and Gram-negative and Gram-positive bacteria were successfully distinguished based on the differences in their lipid profiles. With the same approach, the bacteria could be distinguished in the same category, regardless of whether they were Gram-negative or Gram-positive bacteria (Figure 5b,c). HCA, a grouping method that functions by creating a dendrogram, was also used to investigate if the lipid profiles of bacteria could be used to distinguish different species of bacteria. As shown in Figure 5d, the lipid profiles of closely related bacterial species (Gram-positive bacteria) are clustered closely together, while rather unrelated bacterial species (Gram-negative bacteria) are grouped separately.

To validate the lipid-based results obtained by TD-ESI/MS in this study, MALDI-TOF was used to differentiate the 10 bacterial species based on the differences in protein profiles. The bacterial species on a single colony in the Petri dish were sampled with a metallic inoculating probe first; then, the sample was smeared as a thin film on a MALDI target plate. After an HCCA matrix solution was applied on the smear and dried in the air, the sample spot was analyzed by MALDI-TOF. Figure 6 displays the MALDI mass spectra recorded in the positive mode for the 10 bacterial species. The protein profiles of the Gram-negative bacteria and Gram-positive bacteria were obviously different (compare Figure 6a–e with Figure 6f–j).

Principal component analysis was also used to analyze the protein profiles obtained from the MALDI-TOF analysis. As a result, Gram-negative and Gram-positive bacterial species subjected in this study were successfully distinguished by PCA (Figure 7a). The explained variance contributed by PC1 was about 30% and 25% for PC2. Figure 7b shows the loadings plot of the PCA, which provided information regarding the contribution of individual protein ion signals to variances covered by the respective principal components. For instance, *C. striatum* was located at the lower-left side of the score plot (dark blue spots in Figure 7a), suggesting that the ions (e.g., m/z 2207, m/z 2107, and m/z 2650 in Figure 7b) in the area contributed significantly to the mass spectra of *C. striatum*. By the same token, *S. aureus* was located at the upper-left side of the score plot (orange spots in Figure 7a), suggesting that the ions (eg, m/z 6885, m/z 4306, and m/z 6922 in Figure 7b) in the area contributed significantly to the mass spectra of *S. aureus*. HCA was also used to investigate if the MALDI-TOF protein profiles of the different bacteria followed the bacterial taxonomy as determined by the 16S rRNA gene sequences. As shown in Figure 7c, the protein profiles of closely related bacterial species are clustered closely together, while rather unrelated bacterial species group separately. Results similar to those for the lipid profiles by TD-ESI/MS were obtained.

Owing to the merits of TD-ESI/MS, as mentioned above, the time required for analysis was extremely short, the sampling was simple, the interference of the cultural medium was avoided, and sample pretreatment was unnecessary. Continuous monitoring of the change in the lipid composition in a bacterial colony can be easily achieved for extended periods of bacterial growth time. Figure 8 shows the change in the main lipid ion signals in a colony (*E. coli*) with a culture time of 1–36 h. Due to the reduced oxygen-carrying capacity of the hemoglobin in blood agar, the red color of the blood agar plate became relatively darker than that of the fresh blood (inset of Figure 8a). The TIC of the bacterial lipid ion signals continuously increased with the culture time for the first 8 h (Figure 8a), and then became maximized at 24 h. After 24 h, the TIC of the bacterial lipid ion signals subsequently decreased. Possible reason is that some lipids were switched to other small metabolites (out of detected scan range) after the growth period of bacteria. Figure 8b shows the variation in the three primary lipid ion signals (i.e., m/z 603.5, m/z 563.6, and m/z 589.5) and the three minor lipid ion signals (i.e., m/z 617.5, m/z 631.5, and m/z 549.6) with the culture time (Figure 8b). As can be seen, the ion signal of the primary lipids substantially increased between 8 and 24 h, and continuously decreased after that. On the other hand, the three minor lipid ion signals gradually increased between 8 and 24 h, and then decreased after that.

## 3. Materials and Methods

### 3.1. Chemical Reagents and Materials

The methanol and acetic acid used to generate the electrospray in a TD-ESI source were purchased from Merck (Darmstadt, Germany) and Sigma-Aldrich (St Louis, MO, USA), respectively. The acetonitrile purchased from Merck (Darmstadt, Germany) was used to prepare the MALDI matrix solution. Distilled deionized water (purified with a PURELAB Classic UV from ELGA, Marlow, UK) was used to prepare the electrospray solution. Trifluoroacetic acid (TFA) purchased from Merck (Darmstadt, Germany) was used to hydrolyze the thick peptidoglycan derived from the bacteria. Blood agar plate (BAP) consisted of tryptic soy agar (TSA), and 5% sheep blood was purchased from BD (Franklin Lakes, NJ, USA) and used for the isolation and cultivation of the microorganisms in this study. Alpha-cyano-4-hydroxycinnamic acid (HCCA), a MALDI matrix, was obtained from Sigma Aldrich (St Louis, MO, USA). All the chemicals and solvents were used without further purification.

### 3.2. Culturing of Bacterial Strains

For the optimization experiments, ten different bacterial species including *Escherichia coli* (*E. coli*), *Klebsiella pneumonia* (*K. pneumonia*), *Moraxella catarrhalis* (*M. catarrhalis*), *Pseudomonas aeruginosa* (*P. aeruginosa*), *Serratia marcescens* (*S. marcescens*), *Bacillus subtilis* (*B. subtilis*), *Corynebacterium striatum* (*C. striatum*), *Enterococcus faecalis* (*E. faecalis*), *Listeria monocytogenes* (*L. monocytogenes*), and *Staphylococcus aureus* (*S. aureus*) were studied. *M. catarrhalis*, *S. marcescens*, *C. striatum*, and *L. monocytogenes* with four isolates were collected from different clinical samples in the Clinical Microbiology Laboratory of Kaohsiung Medical University Hospital (Kaohsiung, Taiwan). Other bacterial isolates were collected and processed at the Bioresource Collection and Research Center (Hsinchu, Taiwan) under standard laboratory protocols (Table 1). Isolated strains of the microorganism were grown on a range of solid agar-based media commonly used in clinical microbiology settings. All the isolated strains were incubated at 36 °C in an aerobic incubator for 19 ± 3 h. After single colonies were grown, the chemical compounds in a colony were sampled with a probe and analyzed by TD-ESI/MS.

### 3.3. Thermal Desorption–Electrospray Ionization/Mass Spectrometry (TD-ESI/MS) Analysis and Multivariate Statistical Analyses

The TD-ESI/MS was set up in a similar manner as that described in detail in our previous publication [35]. Briefly, a TD-ESI source comprised of a metallic inoculating probe (60 mm long, 0.2 mm thickness, OD = 2.75 mm, ID = 1.5 mm; Ming Yuh Scientific Instruments Tainan, Taiwan), a thermal desorption unit, and an electrospray ionization interface and was coupled to a linear ion trap mass spectrometer (LTQ XL Thermo Scientific, Waltham, MA, USA). The tip of the metallic inoculating probe was used to collect simply by dipping and removing the metallic inoculating probe on the single colony growing on agar in a Petri dish. For organic solvent extraction, the sample collected on the probe was immersed into a sterilized Eppendorf vial containing 100 µL chloroform/methanol solution (2:1, *v*/*v*) and gently shaken for 5 s. For acid hydrolysis, the sample collected on the probe was immersed in a sterilized Eppendorf vial containing 100 µL 5% TFA solution for 5 s. Liquid samples (ca. 2 µL) were collected by dipping and removing the metallic inoculating probe from the vial. The probe was then inserted in a TD-ESI source for further TD-ESI/MS analyses, and, following this, the metallic inoculating probe was burnt for 5 s with a high-temperature flame by means of a handheld butane torch (K-747, Changhua, Taiwan) to remove any residues. The desorption temperature of the TD-ESI source was set at 280 °C using a temperature controller (ANLY AT-502, Taipei, Taiwan). A metal tubing (OD = 15 μm, ID = 5 μm) was attached to the heated oven in order to carry the heated nitrogen gas prior to entering the desorption area. The heated nitrogen stream (1 L/h) flew from the top of the TD unit down to an ESI plume. The electrospray solution consisted of 50% methanol (v/v) with 0.1% acetic acid. A high voltage (4.0 kV) was applied to the ESI solution in a capillary (OD = 375 μm, ID = 100 μm; Polymicro, Phoenix, AZ) to induce electrospray ionization via solution conduction.

To distinguish different bacterial species, the experimental results obtained from the TD-ESI/MS were further analyzed with PCA and HCA, a commonly used multivariate statistical method that reduces the dimensionality of a dataset while retaining the information present in the original mass spectra. To perform the PCA and HCA analysis, the ion signals detected from the bacterial colony were processed with a MATLAB software tool integrated with the Mass Profiler Professional 13.1 suite (Agilent Technologies Inc. and Strand Life Sciences Pvt. Ltd., Santa Clara, CA, USA).

### 3.4. Matrix-Assisted Laser Desorption Ionization Time-of-Flight Mass Spectrometry (MALDI-TOF/MS) Analysis and Multivariate Statistical Analyses

The HCCA matrix solution (10 mg/mL) was prepared in 50% acetonitrile with 2.5% TFA. The samples collected from a single bacterial colony by dipping and removing the metallic inoculating probe at the colony were smeared as a thin film on a MALDI target plate (Bruker Daltonics, Bremen, Germany). One microliter of HCCA matrix solution was then deposited onto the smear and dried in the air (this HCCA matrix solution is used to assist the desorption and ionization of proteins in MALDI processes and to extract ribosomal proteins from bacterial cells for MALDI-TOF/MS measurement). The target plate was then sent into the ion source of a MALDI-TOF mass spectrometer (AutoFlex III, Bruker Daltonics, Germany), and the sample spot was irradiated with an Nd:YAG laser (355 nm; 3 ns pulse duration) for desorption/ionization of the analytes. The analyte ions were detected with a TOF analyzer operated in a linear mode with spectra recorded over the mass range of m/z 2k to 20k, and the results obtained from 1000 laser shots were averaged to obtain the representative mass spectra. Positive-ion mass spectra were acquired at an acceleration voltage of 19 kV under a delayed extraction mode.

The ion peaks on the MALDI mass spectra were selected with ClinProTools software (v. 2.2, Bruker Daltonics). The functions of this software include ion peak definition (signal-to-noise ratio ≥ 3; relative threshold base peak ≥ 1% in total average spectrum), baseline subtraction (Convex Hull Baseline, 0.8% Baseline Flatness), intensity normalization, peak picking, and statistical testing (Wilcoxon/Kruskal–Wallis and t-test/analysis of variance). In the univariate analysis, variable signals were classified by P values of the Wilcoxon/Kruskal–Wallis test, and the intensities of the signals in the average spectra were normalized in arbitrary units. The resulting peaks on the mass spectrum of each sample were processed using PCA and HCA in the MATLAB software tool integrated in ClinProTools 2.2.

## 4. Conclusions

This research successfully integrated solvent extraction, probe sampling, AIMS, and multivariate statistics to rapidly characterize lipids in a single bacterial colony growing on agar in a Petri dish and distinguished different bacterial species. The results of the differentiation of bacterial species by the differences in their lipid profiles using TD-ESI/MS coupled with PCA/HCA matched well with the results obtained by MALDI-TOF/MS combined with PCA/HCA for the differences in the protein profiles. It is anticipated that with the help of this cutting-edge technology, the normally time-consuming incubation can be performed more efficiently, allowing us to identify microorganisms in a precise and timely manner. The highly efficient analytical platform developed in this study is expected to add a novel diagnostic role to the clinical practice of identifying bacterial species.

## Figures and Tables

**Figure 1 molecules-27-02772-f001:**
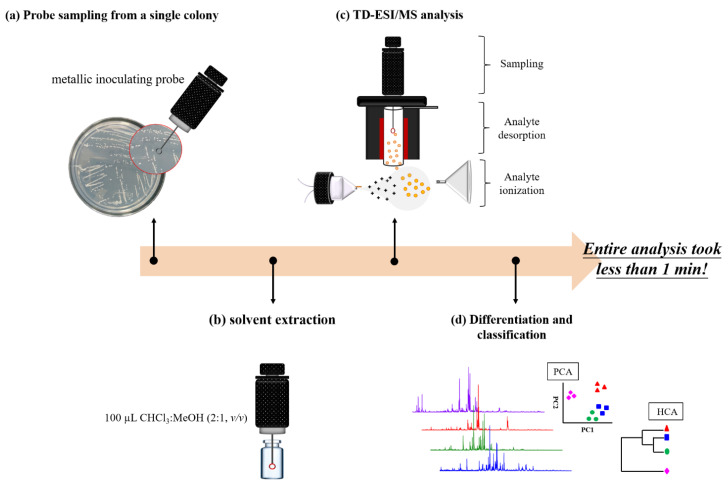
Analytical processes for the direct characterization of microorganisms: (**a**) a metallic inoculating probe was used to collect a single colony of bacterial species on a Petri dish, (**b**) analytes on the probe was extracted with organic solvent; (**c**) analytes on the probe were thermally desorbed and ionized in a TD-ESI source; and (**d**) mass spectral data of the corresponding bacteria were acquired followed by PCA and HCA. The metallic inoculating probe was then cleaned up by a flame from handheld butane torch. The entire analysis took less than 1 min.

**Figure 2 molecules-27-02772-f002:**
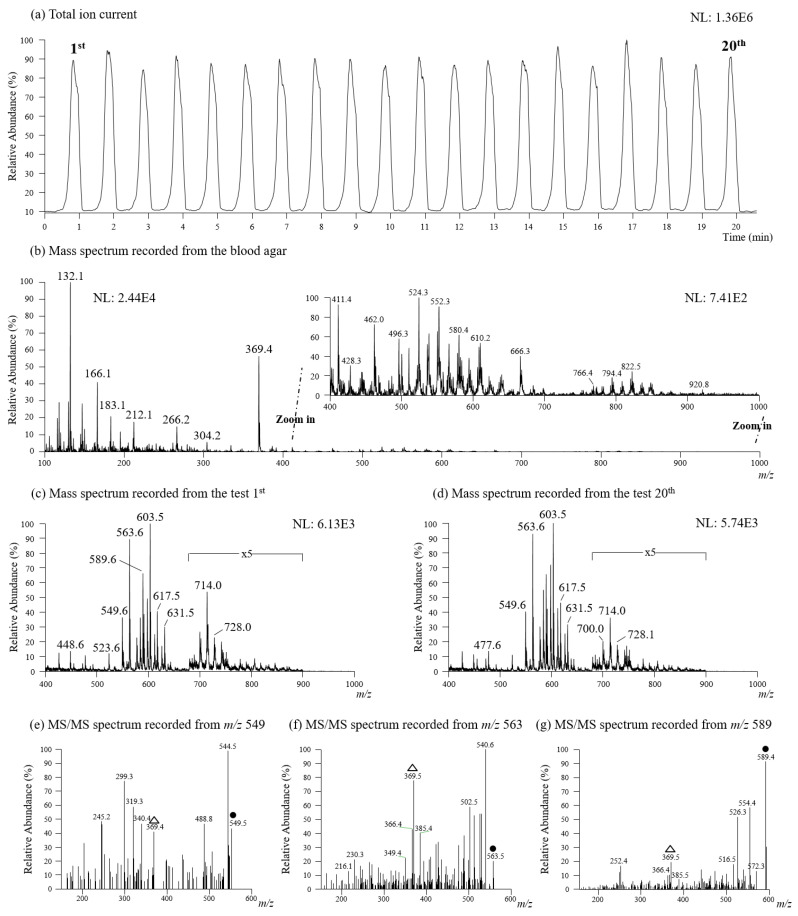
(**a**) Repeatability test (20 determinations) was performed by TD-ESI/MS to directly analyze single colonies of *E. coli*, a Gram-negative bacterium. (**b**) The mass spectrum of blood agar recorded in positive mode. The lipid ions detected from a single colony of *E. coli* with (**c**) the 1st (**d**) the 20th test were detected in positive mode. Product ion mass spectra for (**e**) m/z 549, (**f**) m/z 563, and (**g**) m/z 589 recorded from *E coli*.

**Figure 3 molecules-27-02772-f003:**
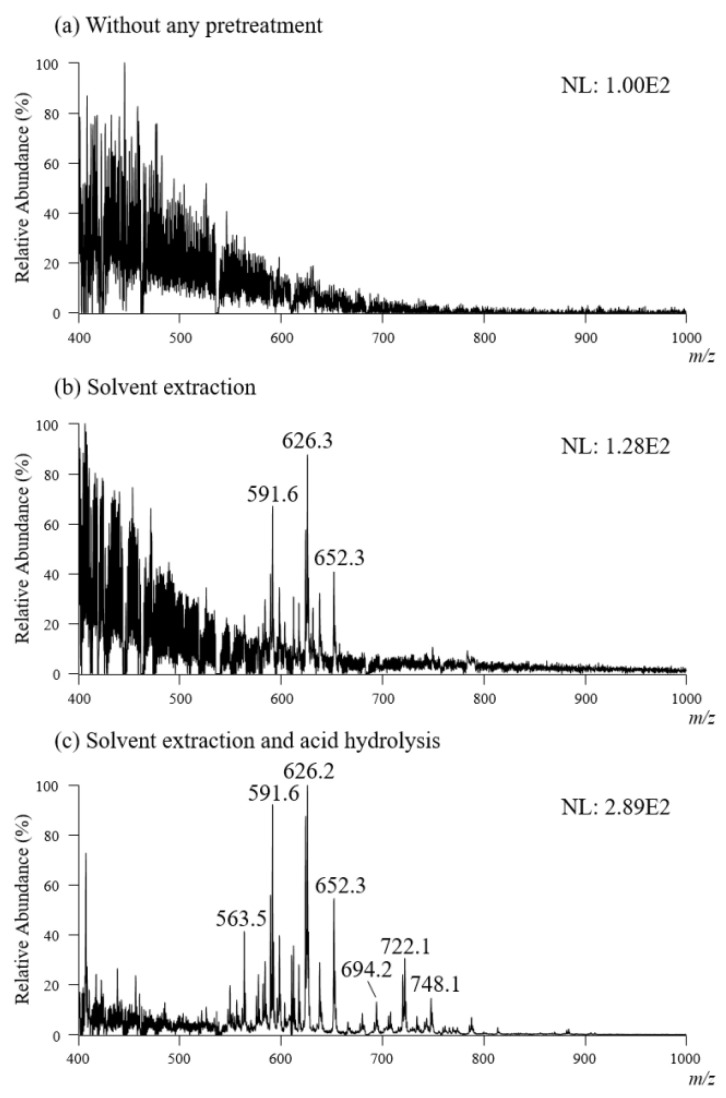
TD-ESI mass spectra recorded in the positive mode of *E. faecalis*: (**a**) without any pretreatment, (**b**) extracted with chloroform/methanol (2:1, *v*/*v*) solution, and (**c**) treated with chloroform/methanol (2:1, *v*/*v*) solution containing 5% TFA.

**Figure 4 molecules-27-02772-f004:**
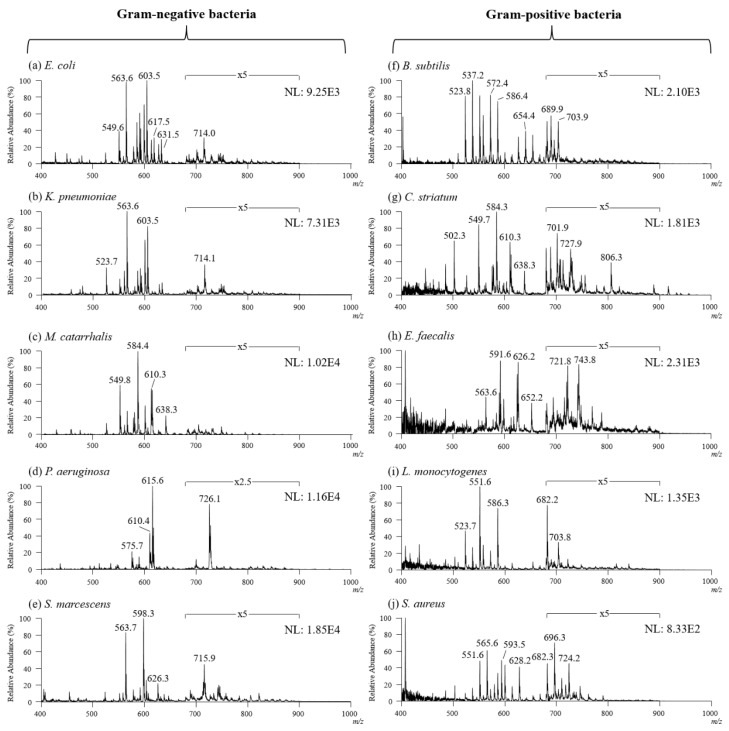
TD-ESI mass spectra recorded in positive mode of Gram-negative bacteria: (**a**) *E. coli*, (**b**) *K. pneumonia*, (**c**) *M. catarrhalis*, (**d**) *P. aeruginosa*, and (**e**) *S. marcescens*. TD-ESI mass spectra recorded in positive mode of Gram-positive bacteria: (**f**) *B. subtilis*, (**g**) *C. striatum*, (**h**) *E. faecalis*, (**i**) *L. monocytogenes*, and (**j**) *S. aureus*.

**Figure 5 molecules-27-02772-f005:**
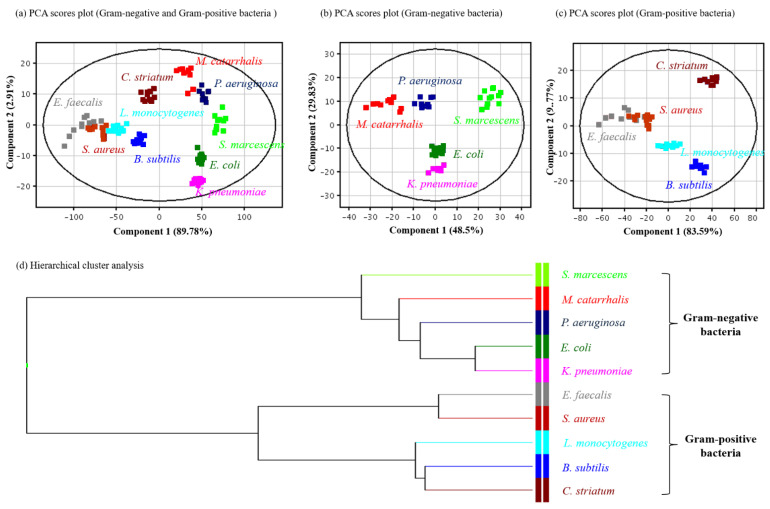
The PCA scores plots (**a**–**c**) and HCA (**d**) of the lipid ion signals on the TD-ESI mass spectra of extracts from 10 bacterial species. Each colony was determined by 10 measurements.

**Figure 6 molecules-27-02772-f006:**
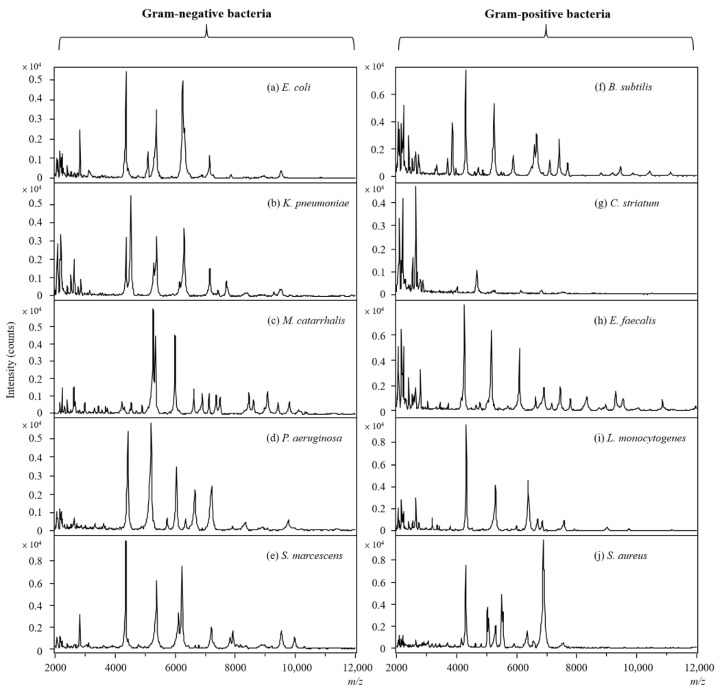
MALDI mass spectra recorded in positive mode for Gram-negative bacteria: (**a**) *E. coli*, (**b**) *K. pneumonia*, (**c**) *M. catarrhalis*, (**d**) *P. aeruginosa*, and (**e**) *S. marcescens*. MALDI mass spectra recorded in positive mode for Gram-positive bacteria: (**f**) *B. subtilis*, (**g**) *C. striatum*, (**h**) *E. faecalis*, (**i**) *L. monocytogenes*, and (**j**) *S. aureus*.

**Figure 7 molecules-27-02772-f007:**
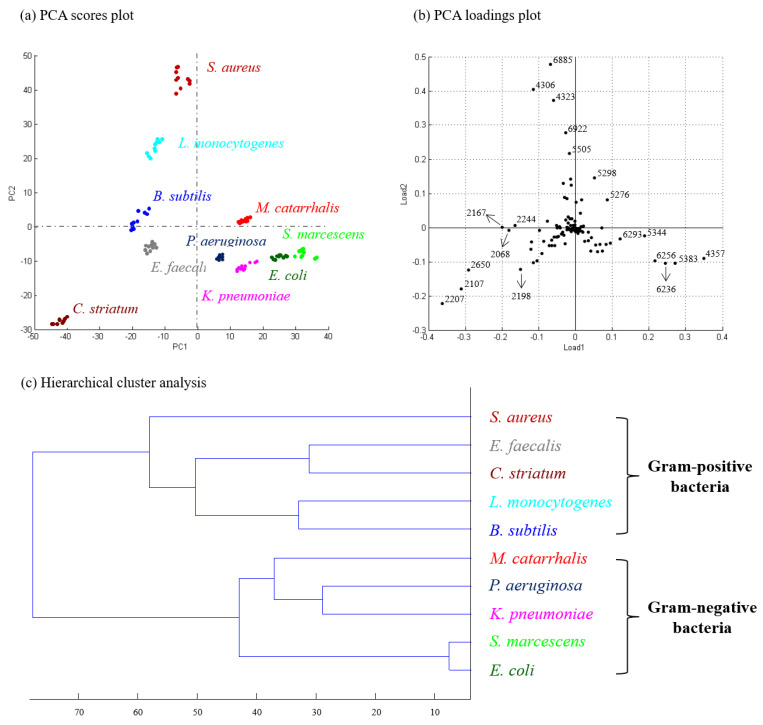
The PCA score plot (**a**), PCA loading plot (**b**), and HCA (**c**) of the protein ion signals on the MALDI-TOF mass spectra of extracts from 10 bacterial species. Each colony was determined by 10 measurements.

**Figure 8 molecules-27-02772-f008:**
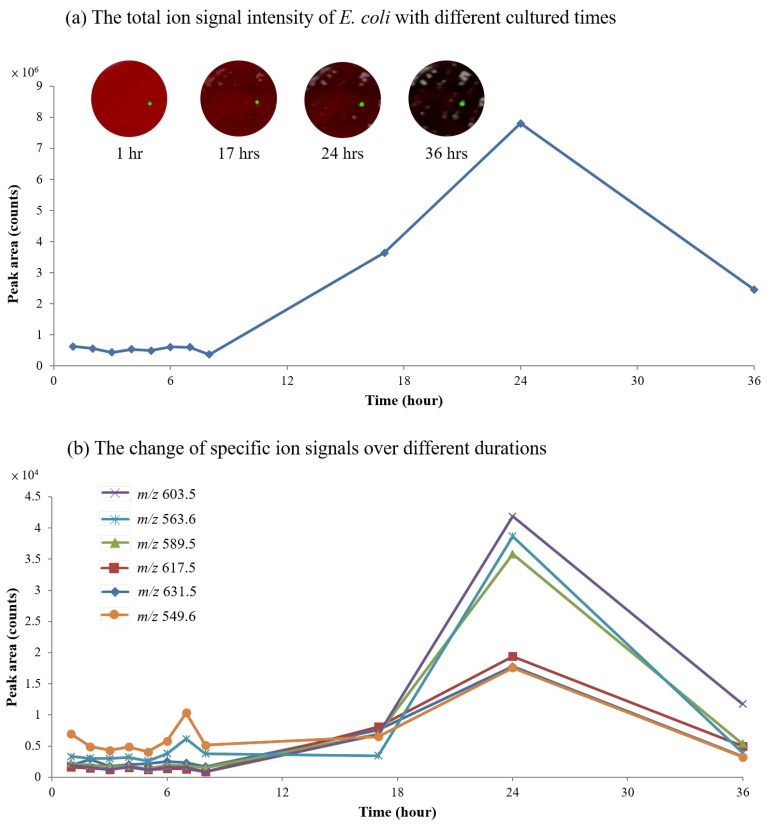
Time-dependent analysis at a single location within an *E. coli* colony indicating (**a**) the correlation between ion signal intensity and incubation time (inset: photos of *E. coli* colony growing on a Petri dish with time; green dot: sampling spot), and (**b**) the changes in the ion signal intensity of specific molecules over different durations using TD-ESI/MS. Note: The Y-axis of the line graph was used as an averaged peak area recorded on triplicate experiments to reduce operative variations.

**Table 1 molecules-27-02772-t001:** Bacterial species tested in this study.

Gram-Negative Bacteria	Gram-Positive Bacteria
*Escherichia coli* (*E. coli*)	*Bacillus subtilis* (*B. subtilis*)
(ATCC 11775)	(ATCC 6051)
*Klebsiella pneumonia* (*K. pneumonia*)	*Corynebacterium striatum* (*C. striatum*)
(ATCC 10031)	(ATCC BAA-1293)
*Moraxella catarrhalis* (*M. catarrhalis*)	*Enterococcus faecalis* (*E. faecalis*)
(ATCC 25238)	(ATCC 29212)
*Pseudomonas aeruginosa* (*P. aeruginosa*)	*Listeria monocytogenes* (*L. monocytogenes*)
(ATCC 10145)	(ATCC 15313)
*Serratia marcescens* (*S. marcescens*)	*Staphylococcus aureus* (*S. aureus*)
(ATCC 8100)	(ATCC 33591)

## Data Availability

The data presented in this study are available in this article.

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
