# Peer review of "Rapid Characterization of Bacterial Lipids with Ambient Ionization Mass Spectrometry for Species Differentiation"

_molecules, 2022, doi:10.3390/molecules27092772_

Round 1

Reviewer 1 Report

Overall impression

The introduction has a good flow and provides a sufficient overview of current methods. However, it is missing a brief (manuscript-scope relevant) description of the bacterial strains selected for the study. For example, are the strains difficult to distinguish one from another? The methods section is informative and contains sufficient information about the applied protocols with two exceptions: (1) there is no information regarding data input for statistical analysis (was peak intensity or m/z used in PCA and HCA?) and (2) the description of the MALDI-TOF experiment does not include how many replicates were averaged per single colony location. The presented spectra are high quality, TD-ESI/MS as well as MALDI-TOF. Some of the results are NOT described (lines 269-271: “Figure 6 displays positive MALDI mass spectra of the 10 bacterial species. The protein profiles between Gram-negative bacteria and Gram-positive bacteria were obviously different”). It would be interesting to compare the profiles obtained in the TD-ESI/MS and MALDI-TOF experiments. In most cases, the presented figures are simple, almost self-explanatory and contain informative, concise figure legends (exception: in Fig5d the two main clusters labels need to be switched). This manuscript provides a sufficient literature review (references list). The Abstract could be more concise and, unfortunately, it misses the significance and novelty of the presented research.

The manuscript was carefully edited before submission. I found only one typo (“if” should be “of”, line 32). However, the entire text needs to be rewritten to fix all grammatical errors. There are so many of them that I will not give any specific examples. In addition, some words/expressions should be replaced/reworded, for example (just a few): “repeatability” for “reproducibility”; “scoring plots” for “score plots”; “sweet spot” (line 99) is jargon; lipids are NOT polar molecules (line 85); “positive mass spectra” for “mass spectra recorded in positive mode” (in figure legends and elsewhere); what are the “lower lipids” (line 293)?; “integrated” (line 398)?; results were further “treated” with PCA and HCA (line 42); What is the meaning of “tedious” (line 86) or “complicated” (line 73) other protocols are rather “complex and time consuming” 

Recommendation

Despite the high quality of the collected data, my recommendation is to reject the manuscript from further review based on following:

  1. Missing experiments (controls)

a) (Major point) There is absolutely NO evidence that presented profiles obtained in TD-ESI/MS are lipid profiles

b) Spectrum of sampled agar from petri dish is missing (background)

  1. Lack of novelty

Bacterial profiling and bacteria identification using mass spectrometry is a well-established platform. Some of the mass spectrometers are even customized to be used in the field (portable, compact size). Some of the models contain a thermal desorption attachment, for example, for quick soil analysis in the field or at the airport to quickly and efficiently screen different surfaces for hazardous materials or illegal drugs.

  1. Lack of (presented) significance (motivation for conducting this research)

One of the mentioned reasons to identify bacterial strains using TD-ESI/MS were:

a) Inexpensive compared to other protocols

Any mass spectrometer is actually more expensive than any thermal cycler

b) Robust and fast

Compounds are analyzed in positive mode only; in addition, without any reference (data base) or compound identification this kind of profiling has little value from either a clinical or non-clinical stand point

c) No sample preparation required

Actually, the gram-positive bacterial samples had to be extracted and hydrolyzed to obtain good quality data; in fact, the presented MALDI-TOF protocol required less preparation

d) Incorrect data interpretation

Lines 287-289: “Due to the reason of nutrient depletion on Petri dish for E. coli (agar color became deep red), the TIC of bacterial lipid ion signals subsequently decreased after that.”

Since the single colony was sampled from the same spot a number of times, the observed decrease in TIC signal is rather related to lack of cells remaining in the sampling spot.

Additional issues to clarify

  1. It is unclear where the “lipids” come from (secondary metabolites, cell membrane, intracellular, agar)
  2. It is unclear how the data presented in Fig.4 were collected (with or without extraction)
  3. It is unclear if lipids from bacteria and mammalian cells are similar or not

Lines 105-110: “However, the similarity between bacteria and mammalian fatty acids may overlap the direct analysis of patient specimens [24]. As generally known, bacteria possess remarkably different phospholipid sets from that of mammalian cells [25,26], so different bacteria species are distinguishable based on the differences of their lipid profiles.”

  1. It is unclear how many times the real-time experiment was repeated (Is Fig.8 prepared based on a single trial?)

Author Response

Please see the attached file, Thank you very much.

Reviewer 2 Report

Hung Su1, et al., present their work, entitled “Rapid Characterization of Bacterial Lipids with Ambient Mass 2 Spectrometry for Species Differentiation”. This is a very interesting study with a wealth of experimental results that collectively create a compelling story.  The methods and analyses appear sound and appropriate, and the data are clearly presented.  In addition to that the answers to the following points and improvements should be considered.

  1. The author should draw a graphical abstract to better understand the readers.
  2. In the introduction, it would be necessary to make a punctual review of the Ambient Mass Spectrometry carried out for bacterial species differentiation with the other species under study, mentioning the other type of bacterial lipids/protein showing the deficiencies and present the purpose of this manuscript by reporting what is new in the present work.
  3. The author should improve the resolution of Figures 1, 2, 4, 5, and 7 because these figures are hazy and image quality should be enhanced.
  4. I suggest that in the result and discussion section author should also include references to studies with similar compositions or with molecules. 
  5. The conclusion section should be revised and concise.

Reviewer 3 Report

The manuscript entitled: "Rapid Characterization of Bacterial Lipids with Ambient Mass 2 Spectrometry for Species Differentiation” presents a study where authors developed a new methodology for bacterial differentiation, based on mass spectrometry technique. The authors evaluated the use of thermal desorption-electrospray ionization/mass spectrometry (TD-35 ESI/MS) and multivariate statistical analysis to rapidly differentiate bacterial species 36 based on the differences in their lipid profiles. The presented study is interesting, the developed methodology seems to be simple and high throughput to the study aim.
In my point of view, the manuscript is well written and presents a promising tool for the microbiology science field. I did not have any concerns or doubts about the submitted manuscript. Thus, I recommend the acceptance of the present manuscript as it is presented.

Author Response

This study will undertake in-depth research on the reviewer's positive comments. Thank you very much.

Round 2

Reviewer 1 Report

Thank you for addressing most of the questions and concerns. Please see below my further comments.

A) Comment1: Missing experiments (controls)

There is absolutely NO evidence that presented profiles obtained in TD-ESI/MS are lipid profiles

Answer1: Earlier literature has reported that these ion signals are mainly from phosphatidylglycerols (PGs), phosphatidylethanolamines (PEs), and phosphatidic acids (PAs) [Refs 33-35]. It has to be noted that the analytical techniques used in these studies are either GC/MS or LC/MS/MS [Refs 50,51]. (Please see page 8, line 264-268) More information has been added in the revised manuscript. The mass spectrum of sampled agar has been added in the revised manuscript. (Please see Fig. 2b)

Comment2: This is an absolutely critical piece of information since this is a “proof of concept”.

Lipids have a very characteristic fragmentation pattern. Recording ms/ms data for a few selected ions would be enough proof (either on Bruker or Thermo instruments).

B) One of the motivations to develop new protocols/methods was the capability of unambiguously identifying bacterial species.

“Therefore, despite its role as the gold standard, not all bacterial species can be confidently identified by the difference of their 16S rRNA sequence. This makes the development of other identification approaches necessary [9,10]. Additionally, the genotypic approach generally needs complex and time-consuming sample preparation, is comparably expensive and requires at least several hours for identification.”; lines 70-74, also see cover letter “lack of novelty”).

Provided protocol does not offer such an option. It only reveals the difference in membrane composition among known bacterial species. To be able to identify bacterial species it would require a standard (reference database) or specific marker which is characteristic only to one species. Following this though there is still no justification for selecting a particular set of gram-negative/positive bacteria.

C) Comment1: Incorrect data interpretation Lines 287-289: “Due to the reason of nutrient depletion on Petri dish for E. coli (agar color became deep red), the TIC of bacterial lipid ion signals subsequently decreased after that.” Since the single colony was sampled from the same spot a number of times, the observed decrease in TIC signal is rather related to lack of cells remaining in the sampling spot.

Answer1: Thank you for the reviewer’s reminder. The correct interpretation has been added to the revised manuscript. (Please see page 12, line 338-344)

Comment2: Since cell depletion during multiple sampling of one spot is an obvious drawback of the TD-ESI/MS, how reliable can the obtained real-time data be?

D) Comment1: It is unclear how many times the real-time experiment was repeated (Is Fig.8 prepared based on a single trial?)

Answer1: As shown in Fig. 8, the Y-axis of the line graph was used averaged peak area recorded on triplicate experiments to reduce operative variations. More information was added in the revised manuscript. (Please see page 13, line 355-356)

Comment2: Error bars should be added to the plot or the legend should comment about standard deviation between measurements.

Author Response

Thank you for your critical comment. We have considered the reviewer’s comments and suggestions and appropriate changes have been made in the revised manuscript
